# Inconsistent PCR detection of Shiga toxin-producing Escherichia coli: Insights from whole genome sequence analyses

**Vinicius Silva Castro**[1,2,3], **Rodrigo Ortega Polo**[4], **Eduardo Eustáquio de Souza Figueiredo**[2], **Emmanuel Wihkochombom Bumunange**[5], **Tim McAllister**[4], **Robin King**[6], **Carlos Adam Conte-Junior**[1], **Kim Stanford**[3]*

1 Institute of Chemistry, Federal University of Rio de Janeiro, Rio de Janeiro, Brazil, 2 Department of Food and Nutrition, Federal University of Mato Grosso, Cuiaba, Brazil, 3 Department of Biological Sciences, University of Lethbridge, Lethbridge, Canada, 4 Lethbridge Research and Development Centre, Agriculture and Agri-Food Canada, Lethbridge, Canada, 5 Department of Ecosystem and Public Health, Faculty of Veterinary Medicine, University of Calgary, Calgary, Canada, 6 Alberta Agriculture and Forestry, Edmonton, Canada

* kim.stanford@uleth.ca

**Data Availability Statement:** All sequence data have been uploaded on NCBI database and are

## Abstract

Shiga toxin-producing Escherichia coli (STEC) have been linked to food-borne disease outbreaks. As PCR is routinely used to screen foods for STEC, it is important that factors leading to inconsistent detection of STEC by PCR are understood. This study used whole genome sequencing (WGS) to investigate causes of inconsistent PCR detection of $stx_1$, $stx_2$, and serogroup-specific genes. Fifty strains isolated from Alberta feedlot cattle from three different studies were selected with inconsistent or consistent detection of stx and serogroup by PCR. All isolates were initially classified as STEC by PCR. Sequencing was performed using Illumina MiSeq® with sample library by Nextera XT. Virtual PCRs were performed using Geneious and bacteriophage content was determined using PHASTER. Sequencing coverage ranged from 47 to 102x, averaging 74x, with sequences deposited in the NCBI database. Eleven strains were confirmed by WGS as STEC having complete stxA and stxB subunits. However, truncated stx fragments occurred in twenty-two other isolates, some having multiple stx fragments in the genome. Isolates with complete stx by WGS had consistent $stx_1$ and $stx_2$ detection by PCR, although one also having a $stx_2$ fragment had inconsistent $stx_2$ PCR. For all STEC and 18/39 non-STEC, serogroups determined by PCR agreed with those determined by WGS. An additional three WGS serotypes were inconclusive and two isolates were Citrobacter spp. Results demonstrate that stx fragments associated with stx-carrying bacteriophages in the E. coli genome may contribute to inconsistent detection of $stx_1$ and $stx_2$ by PCR. Fourteen isolates had integrated stx bacteriophage but lacked complete or fragmentary stx possibly due to partial bacteriophage excision after subcultivation or other unclear mechanisms. The majority of STEC isolates (7/11) did not have identifiable bacteriophage DNA in the contig(s) where stx was located, likely increasing the stability of stx in the bacterial genome and its detection by PCR.

included in BioProject # PRJNA601484 https://www.ncbi.nlm.nih.gov/bioproject/PRJNA601484.

**Funding:** CAPES/Brazil Visiting Professor (Process: PVEX 88881.169965/2018-01). Fund Carlos Chagas Filho de Amparo a Pesquisa do Estado do Rio de Janeiro (Process: E-26/201.859/2019). Canada Alberta Project (CAP) Accelerating the Advancement of Agriculture AGUCMINT 5581487. Results-Driven Agriculture Research (RDAR) grant 2021R010R The funders had no role in study design, data collection and analysis, decision to publish, or preparation of the manuscript.

**Competing interests:** The authors have declared that no competing interests exist.

## 1. Introduction

Shiga toxin-producing *Escherichia coli* (STEC) is one of the most important pathogens in food-borne illness. Currently, STEC includes more than 400 strains, with O157 and the non-O157 "big six" (O26, O45, O103, O111, O121, and O145) serogroups being most frequently linked to hemorrhagic colitis in humans [1]. However, due to low cell numbers to trigger an infection and the diversity of STEC it can be challenging to isolate or identify specific serogroups associated with contaminated foods.

Several methodologies have been used to identify or isolate STEC including immunomagnetic separation (IMS), a selective and enriched media, PCR, and qPCR [2–7]. However, there is still a lack of a gold standard methodology for isolating STEC [8]. Also, the development of specific methods according to the sample matrix could increase sensitivity and lower the threshold of detection of STEC strains. To further these aims, antimicrobials are commonly added to STEC media to prevent plate overgrowth [9], but this practice does not guarantee that only STEC will be isolated, or discriminate STEC serogroups.

For identification of STEC strains, PCR reactions are commonly based on the presence of Shiga toxin genes and can also be applied to determine bacterial serogroup through the amplification of genes responsible for the synthesis of O-antigens (*wzx* and *wzy;* [10–12]). Factors such as the presence of bacteriophage (phage) which are not incorporated into the bacterial genome and DNA purity can influence the accuracy and sensitivity of detecting STEC using PCR [13, 14]. Furthermore, repeated subculturing of STEC can result in the loss of *stx*-coding phage [15], even with the first subculture [16]. Moreover, in a recent study Macori et al. [17] observed that qPCR amplified free phages encoding *stx* in samples collected from the rectal anal junction of sheep. Accordingly, there is growing consensus that more investigation is needed to evaluate the impact of *stx*-carrying free-phages or integration and loss of *stx*-phages from bacterial genomes on the detection and confirmation of STEC, as false-positive (PCR-positive but no *stx* integrated into genome) or false-negative (PCR negative but with *stx* present) results have consequences for food safety.

This study used whole genome sequencing (WGS) of *E. coli* isolated from feces of western-Canadian cattle to: (i) compare whole genome sequences with previous PCR detection of Shiga toxins and serogroup; (ii) investigate the presence and heterogeneity of *stx*-encoding phages; and (iii) determine the presence of other virulence factors and antimicrobial resistance of isolates.

## 2. Material and methods

### 2.1 Bacterial strains and culture

A total of fifty *E. coli* previously isolated from cattle feces in three different studies were used for WGS and all strains were encoded with the acronym CAP due to financial support of the Canadian Agricultural Partnership. Forty-eight strains were isolated from feces of western-Canadian slaughter cattle collected from the floor of transport trailers [18], one strain was isolated from the pen floor of an Alberta feedlot [19], and one was isolated in feedlot cattle feces in 2017 [20]. Isolates were selected for WGS based on consistent or inconsistent PCR detection of *stx₁* and/or *stx₂* and/or serogroup from 750 strains analysed by Zhang et al. [21] and belong to a larger pool of approximately 15,000 isolates [20].

### 2.2 PCR

Primers designed by Conrad et al. [10] were used for detection of *stx₁* and *stx₂* (Table 1). The reactions were performed as follows: 95˚C for 5 min, followed by 35 cycles of 94˚C for 30 s,

**Table 1. Primers used to detect *stx* and serogroup.**

| *Stx* | Reference | Primer | Sequence (5′ - 3′) | Amplicon size |
|---|---|---|---|---|
| | Conrad et al. 2014 | $stx_1$ | GGATGATCTCAGTGGGCGTTGATGCCATTCTGGCAACTCG | 216 |
| | Conrad et al. 2014 | $stx_2$ | ACTGTCTGAAACTGCTCCTGTGCGCTGCAGCTGTATTACTTTCC | 307 |
| | Scheutz et al. 2012 | $stx_1$-det-F1 | GTACGGGGATGCAGATAAATCGC | 209 |
| | Scheutz et al. 2012 | $stx_1$-det-R1 | AGCAGTCATTACATAAGAACGYCCACT | |
| | Scheutz et al. 2012 | F4 ($stx_2$) | GGCACTGTCTGAAACTGCTCCTGT | 627 |
| | Scheutz et al. 2012 | R1 ($stx_2$) | ATTAAACTGCACTTCAGCAAATCC | |
| | Scheutz et al. 2012 | F4-f ($stx_2$) | CGCTGTCTGAGGCATCTCCGCT | 625 |
| | Scheutz et al. 2012 | R1-e/f ($stx_2$) | TAAACTTCACCTGGGCAAAGCC | |
| Serogroup | Conrad et al. 2014 | O157 | GGCTGGGAATGCATCGGCCTTGTCAGAGCAGCACCAAGACTGG | 1083 |
| | Conrad et al. 2014 | O26 | ATTGCAGCGCCTATTTCAGCATTAGAAGCGCGTTCATCCCT | 200 |
| | Conrad et al. 2014 | O45 | GATCTGTGGAGCCGAGATGGTTTGAGACGAGCCTGGCTTT | 250 |
| | Conrad et al. 2014 | O103 | ATCTTCTTGCGGCTGCAGTTAAAGGCGCATTAGTGTCTGC | 340 |
| | Conrad et al. 2014 | O121 | GGTTGGATGGGTGGAACCTTAGCAAGCCAAAACACTCAACA | 595 |
| | Conrad et al. 2014 | O145 | GCGGGTGTTGCCCGTTCTGTACGGCATTCCGCTGCGAGTT | 766 |

60˚C for 45 s, 72˚C for 90 s, and a final extension of 72˚C for 5 min. Conrad et al. [10] primers were also used for detection of serogroups (O26, O45, O103, O121, O145 O157; Table 1). PCRs contained a final volume of 25µL and 0.2 µM each primer, 1x HotStar Taq Plus Master-Mix (Qiagen® Hilden, Germany), 1x Coral Load PCR buffer, 2 µL DNA template, and nuclease-free water. The reactions were performed in a Veriti™ Dx Thermal Cycler (Applied Biosystems). To ensure that the PCR primers used were not responsible for inconsistent $stx_1$ and $stx_2$ results, virtual PCR was performed for the 50 isolates using Geneious 10.2.6 software (Biomatters, Auckland, Australia) to compare primers of Scheutz et al. [22] and Conrad et al. ([10]; Table 1). Also, two base pair (bp) mismatches between primer and sequences for both *stx* and serogroup were allowed to ensure that inconsistences which can lead to amplification were considered. For other configurations default parameters were used.

## 2.3 DNA extraction and WGS

Genomic DNA was extracted from overnight bacterial cultures prepared in Luria-Bertani broth (Merck, Darmstadt, Germany) using the ZR Fungal/Bacterial DNA MiniPrep™ kit (Epigenetics Company, Irvine, CA, USA) according to the manufacturer's instructions. DNA was quality checked and quantified using a Qubit fluorimeter (ThermoFisher, Waltham, MA, USA) and a TapeStation 4200 system (Agilent, Santa Clara, CA, USA). Sample libraries were prepared using the Nextera XT library preparation kit protocol (Illumina, Inc., San Diego, CA, USA). Sequencing was performed on the Illumina MiSeq platform using the MiSeq Reagent Kit V2 to produce 251 bp paired-end reads. Sequencing was performed at the Agri-Food Laboratories, (Alberta Agriculture and Forestry, Edmonton, AB, Canada).

## 2.4 Sequencing analysis

Sequencing reads were *de novo* assembled into contigs using the Shovill pipeline (https://github.com/tseemann/shovill). Shovill included trimming, which was performed with Trimmomatic 0.39, and *de novo* assembly was performed with SPAdes version 3.13.1. [23]. Draft genome assemblies were annotated with Prokka [24], included in the NCBI database (BioProject: PRJNA601484), and published by Castro et al. [20]. Sequencing coverage ranged from 47 to 102x, with an average coverage of approximately 74x. A FastQC was applied to all strains to guarantee a

good depth of coverage in each isolate. In addition, contigs were searched against databases for virulence genes (VirFinder; [25]), antimicrobial resistance genes, and plasmids (PlasmidFinder) using ABRicate version 0.8.7 (https://github.com/tseemann/ABRICATE ). Non-O157 *E. coli* serotype determinants (O- and H-antigen sequences) were inferred *in silico* using the EcOH database [26], originally developed for Short Read Sequence Typing for Bacterial Pathogens (SRST2; [26]). The EcOH database contained sequences of O-antigen loci [either *wzx* (O-antigen flippase) and *wzy* (O-antigen polymerase)], or the ABC transporter (*wzm* and *wzt*) and H-antigen (*fliC* and *flnA*) with referenced loci to *E. coli* O-groups and H-types. The virulence factor (VF) profile was generated by searching contigs against the *E. coli*_VF database [27]. Nucleotide sequence identity above 70% to the appropriate reference gene was considered to represent virulence factors. Antimicrobial resistance gene profiles were generated by searching contigs against the Comprehensive Antibiotic Resistance Database [28], and plasmid search profiles were generated by searching contigs against the replicon sequences from the plasmidFinder database [29]. Replicon sequence identity above 80% was used to designate targets as being present in a genome.

Presence of phage sequences in bacterial genomes was assessed using phaster.ca [30, 31]. Phage sequences were compared with reference *stx* genes (NC_004913.3; NC_049944.1; NC_008464.1) using the Blastn platform (NCBI) and to our WGS strains using Geneious Prime (Biomatters, Auckland, NZ). The MAFFT 7.450 tool [32] was used to align *stx* sequence data with that of *stx*-encoding phages obtained from NCBI database using a scoring matrix 200PAM / K = 2, GAP open penalty of 1.53, offset value of 0.123 and automatic determination of sequence direction. The integrity of *stx* (%) was then calculated automatically in the aligned sequences, selecting only bases with agreement between NCBI phage and strain sequences. A heatmap illustrating the presence of phages in bacterial sequences was prepared using Graph-Pad Prism 5.01 (GraphPad Software, San Diego, CA).

## 3. Results

### 3.1 Overall concordance of PCR and WGS

After WGS of the 50 isolates, forty-eight were confirmed as *Escherichia coli* and two (CAP 7, CAP 50) were identified as *Citrobacter* spp. and were removed from further analysis. Within the forty-eight isolates of *E. coli*, only eleven were classified as STEC by WGS [20] as they had contiguous *stx*A and *stx*B subunits forming complete sequences for $stx_1$ or $stx_2$, even though $stx_1$ or $stx_2$ were detected by PCR at least once in all isolates (Table 2). All isolates confirmed as STEC by WGS were also consistently classified as STEC by PCR. To evaluate the effectiveness of the PCR primers, a virtual PCR and a Blastn using the NCBI platform were performed to compare binding of $stx_1$ and $stx_2$ primers to generic *E. coli* (without *stx* presence as determined by WGS) and STEC. Importantly, all STEC confirmed by WGS were positive for $stx_1$ and complete $stx_2$ sequences were found in two STEC (Table 2).

Blastn results showed no $stx_1$ or $stx_2$ primer binding in strains classified as generic *E. coli* by WGS (Table 2). Also, Blastn results discard amplification with other genome sequences, and for isolates not confirmed to be STEC by WGS, the highest score (correspondence between bases of the sequence with the primer) for $stx_1$ was 28.2 (binding of 14 bases of DNA into 25 bases of the forward primer) and for $stx_2$ 30.2 (binding of 15 bases of DNA sequence into 24 bases of the forward primer). Moreover, virtual PCR using the Conrad et al. [10] and Scheutz et al. [22] primers for $stx_1$ and $stx_2$ also indicated amplification only in STEC strains confirmed by WGS.

### 3.2 Primers and phages

Of 48 strains confirmed as *Escherichia coli* by WGS, 10 STEC and 22 non-STEC had up to six *stx*-encoding phages integrated within their bacterial genome (Table 3). For these thirty-two

**Table 2. Previous PCR data for Shiga toxins by conventional and molecular methods and WGS data.**

| Strain | PCR for *stx* (1° assay)[1] | PCR for *stx* (2° assay)[1] | *Stx* fragments by WGS[2] | Blastn for primers and DNA sequence[3] | Virtual PCR using Geneious[4] | Detection of *stx* by WGS |
|---|---|---|---|---|---|---|
| CAP 01 | $stx_1$ | . | $stx_1$ | $stx_1$–28.2 (14/25) $stx_2$–28.2 (14/24) | . | . |
| CAP 02 | $stx_1$ | $stx_1$ | $stx_1$ | $stx_1$–50.1 (25/25) $stx_2$–28.2 (14/24) | $stx_1$ | $stx_{1a}$, $stx_{1b}$ |
| CAP 03 | $stx_1$, $stx_2$ | $stx_1$, $stx_2$ | $stx_1$ | $stx_1$–50.1 (25/25) $stx_2$–48.1 (24/24) | $stx_1$, $stx_2$ | $stx_{1a}$, $stx_{1b}$, $stx_{2a}$, $stx_{2b}$ |
| CAP 04 | $stx_1$ | $stx_1$ | $stx_1$ | $stx_1$–50.1 (25/25) $stx_2$–28.2 (14/24) | $stx_1$ | $stx_{1a}$, $stx_{1b}$ |
| CAP 05 | $stx_1$ | . | $stx_1$ | $stx_1$–26.3 (13/25) $stx_2$–28.2 (14/24) | . | . |
| CAP 06 | $stx_1$ | . | $stx_1$ | $stx_1$–28.2 (14/25) $stx_2$–28.2 (14/24) | . | . |
| CAP 08 | $stx_1$ | . | $stx_1$ | $stx_1$–28.2 (14/25) $stx_2$–28.2 (14/24) | . | . |
| CAP 09 | $stx_2$. | $stx_2$. | . | $stx_1$–28.2 (14/25) $stx_2$–28.2 (14/24) | . | . |
| CAP 10 | $stx_1$ | $stx_1$ | $stx_1$, $stx_2$ | $stx_1$–50.1 (25/25) $stx_2$–28.2 (14/24) | $stx_1$ | $stx_{1a}$, $stx_{1b}$ |
| CAP 11 | $stx_1$ | . | $stx_1$, $stx_2$ | $stx_1$–28.2 (14/25) $stx_2$–28.2 (14/24) | . | . |
| CAP 12 | $stx_2$ | . | . | $stx_1$–28.2 (14/25) $stx_2$–28.2 (14/24) | . | . |
| CAP 13 | $stx_1$ | $stx_1$ | . | $stx_1$–28.2 (14/25) $stx_2$–28.2 (14/24) | . | . |
| CAP 14 | $stx_1$ | $stx_1$, $stx_2$ | $stx_1$, $stx_2$ | $stx_1$–28.2 (14/25) $stx_2$–28.2 (14/24) | . | . |
| CAP 15 | $stx_1$ | . | $stx_1$ | $stx_1$–28.2 (14/25) $stx_2$–28.2 (14/24) | . | - |
| CAP 16 | $stx_1$ | $stx_1$ | $stx_1$ | $stx_1$–50.1 (25/25) $stx_2$–28.2 (14/24) | $stx_1$ | $stx_{1a}$, $stx_{1b}$ |
| CAP 17 | $stx_1$ | $stx_1$ | $stx_1$ | $stx_1$–28.2 (14/25) $stx_2$–28.2 (14/24) | . | . |
| CAP 18 | $stx_1$, $stx_2$ | $stx_1$, $stx_2$ | $stx_1$ | $stx_1$–50.1 (25/25) $stx_2$–48.1 (24/24) | $stx_1$, $stx_2$ | $stx_{1a}$, $stx_{1b}$, $stx_{2a}$, $stx_{2b}$ |
| CAP 19 | $stx_1$ | $stx_1$, $stx_2$ | . | $stx_1$–50.1 (25/25) $stx_2$–28.2 (14/24) | $stx_1$ | $stx_{1a}$, $stx_{1b}$ |
| CAP 20 | $stx_1$ | $stx_1$ | $stx_1$ | $stx_1$–28.2 (14/25) $stx_2$–28.2 (14/24) | . | . |
| CAP 21 | $stx_1$, $stx_2$ | . | . | $stx_1$–28.2 (14/25) $stx_2$–28.2 (14/24) | . | . |
| CAP 22 | $stx_1$, $stx_2$ | . | $stx_1$ | $stx_1$–28.2 (14/25) $stx_2$–28.2 (14/24) | . | . |
| CAP 23 | $stx_1$, $stx_2$ | $stx_1$ | $stx_1$ | $stx_1$–50.1 (25/25) $stx_2$–28.2 (14/24) | $stx_1$ | $stx_{1a}$, $stx_{1b}$ |
| CAP 24 | $stx_1$, $stx_2$ | . | . | $stx_1$–26.3 (13/25) $stx_2$–28.2 (14/24) | . | . |
| CAP 25 | $stx_1$ | $stx_1$, $stx_2$ | $stx_1$ | $stx_1$–28.2 (14/25) $stx_2$–28.2 (14/24) | . | . |
| CAP 26 | $stx_1$ | $stx_1$ | $stx_1$ | $stx_1$–28.2 (14/25) $stx_2$–28.2 (14/24) | . | . |
| CAP 27 | $stx_1$ | $stx_1$ | . | $stx_1$–28.2 (14/25) $stx_2$–28.2 (14/24) | . | . |
| CAP 28 | $stx_1$, $stx_2$ | $stx_1$ | . | $stx_1$–28.2 (14/25) $stx_2$–30.2 (15/24) | . | . |
| CAP 29 | $stx_2$ | $stx_1$ | . | $stx_1$–28.2 (14/25) $stx_2$–28.2 (14/24) | . | . |
| CAP 30 | $stx_2$ | $stx_1$ | $stx_1$ | $stx_1$–28.2 (14/25) $stx_2$–28.2 (14/24) | . | . |
| CAP 31 | $stx_1$ | $stx_1$, $stx_2$ | . | $stx_1$–28.2 (14/25) $stx_2$–28.2 (14/24) | . | . |
| CAP 32 | $stx_1$ | $stx_1$, $stx_2$ | $stx_2$ | $stx_1$–50.1 (25/25) $stx_2$–28.2 (14/24) | $stx_1$ | $stx_{1a}$, $stx_{1b}$ |
| CAP 33 | $stx_1$ | $stx_1$ | $stx_2$ | $stx_1$–50.1 (25/25) $stx_2$–28.2 (14/24) | $stx_1$ | $stx_{1a}$, $stx_{1b}$ |
| CAP 34 | $stx_1$, $stx_2$ | $stx_2$ | $stx_1$ | $stx_1$–26.3 (13/25) $stx_2$–28.2 (14/24) | . | . |
| CAP 35 | $stx_1$, $stx_2$ | . | $stx_1$ | $stx_1$–28.2 (14/25) $stx_2$–28.2 (14/24) | . | . |
| CAP 36 | $stx_1$ | $stx_2$ | . | $stx_1$–26.3 (13/25) $stx_2$–28.2 (14/24) | . | . |
| CAP 37 | $stx_1$, $stx_2$ | . | . | $stx_1$–28.2 (14/25) $stx_2$–28.2 (14/24) | . | . |
| CAP 38 | $stx_1$, $stx_2$ | $stx_1$ | . | $stx_1$–28.2 (14/25) $stx_2$–28.2 (14/24) | . | . |
| CAP 39 | $stx_1$, $stx_2$ | $stx_1$, $stx_2$ | $stx_1$ | $stx_1$–28.2 (14/25) $stx_2$–28.2 (14/24) | . | . |
| CAP 40 | $stx_1$ | $stx_1$ | . | $stx_1$–28.2 (14/25) $stx_2$–28.2 (14/24) | . | . |
| CAP 41 | $stx_2$ | $stx_1$, $stx_2$ | . | $stx_1$–28.2 (14/25) $stx_2$–28.2 (14/24) | . | . |
| CAP 42 | $stx_1$ | $stx_1$ | $stx_1$ | $stx_1$–28.2 (14/25) $stx_2$–28.2 (14/24) | . | . |
| CAP 43 | $stx_1$ | $stx_1$ | . | $stx_1$–28.2 (14/25) $stx_2$–28.2 (14/24) | . | . |
| CAP 44 | $stx_1$ | $stx_1$ | $stx_1$ | $stx_1$–28.2 (14/25) $stx_2$–28.2 (14/24) | . | . |
| CAP 45 | $stx_1$ | . | $stx_1$ | $stx_1$–26.3 (13/25) $stx_2$–28.2 (14/24) | . | . |
| CAP 46 | $stx_1$ | $stx_1$ | $stx_1$ | $stx_1$–28.2 (14/25) $stx_2$–28.2 (14/24) | . | . |
| CAP 47 | $stx_1$ | $stx_1$ | . | $stx_1$–50.1 (25/25) $stx_2$–28.2 (14/24) | $stx_1$ | $stx_{1a}$, $stx_{1b}$ |

*(Continued)*

**Table 2.** (Continued)

| Strain | PCR for $stx$ (1° assay)[1] | PCR for $stx$ (2° assay)[1] | $Stx$ fragments by WGS[2] | Blastn for primers and DNA sequence[3] | Virtual PCR using Geneious[4] | Detection of $stx$ by WGS |
|---|---|---|---|---|---|---|
| CAP 48 | $stx_1$ | $stx_1$ | $stx_1$ | $stx_1$−28.2 (14/25) $stx_2$−28.2 (14/24) | . | . |
| CAP 49 | $stx_1$ | . | $stx_1$ | $stx_1$−28.2 (14/25) $stx_2$−28.2 (14/24) | . | . |

[1] Conrad et al. (2014) primers;

[2] Truncated $stxA$ and $stxB$ subunits including some base pair mismatches.

[3] Match of DNA nucleotides and Conrad et al. (2014) primers;

[4] Using both Conrad et al. (2014) and Scheutz et al. (2012) primers. No differences in detection by primer sets.

isolates, up to three fragments of $stx$ (truncated $stxA$ and $stxB$ subunits) were associated with phage DNA insertions (Table 3 and Fig 1). However, presence of $stx$-phages did not guarantee presence of even fragmentary $stx$ and fourteen of the integrated $stx$-phages lacked $stx$ coding sequences. Only one STEC strain confirmed by WGS (CAP 19) did not contain sequences attributed to a $stx$-encoding phage.

One STEC strain with inconsistent PCR detection of $stx_2$ (CAP 32) was found to have a fragment of $stx_2$ integrated in the genome (Table 2). Twenty-two strains classified as generic $E. coli$ by WGS had phage fragments of $stx_1$, and in one case $stx_1$ and $stx_2$, which may have contributed to inconsistent PCR detection of these genes. However, 15 strains previously PCR-positive for $stx_1$ or $stx_2$ lacked $stx$ fragments in their genome and were not confirmed as STEC by WGS. As well, for two isolates even though $stx_2$ fragments were present, $stx_2$ was never detected by PCR. Integrity of $stx$ present in fragments varied from three to 38.7% (Table 3).

$Stx$ phage fragments present in our isolates were compared to phage reference sequences from NCBI and we also performed virtual PCRs using primers designed by Conrad et al. [10] and Scheutz et al. [22]. Virtual PCR results emphasize that all lysogenic phages had insertion locations which corresponded to reference sequences which would have been amplified by both sets of primers. However, no phage sequences were complete as compared to reference sequences, with phage integrity ranging from 1–60% (Table 3). Additionally, there was no difference between the two primer sets [10, 22] in detection of $stx_1$ or $stx_2$ in reference phages.

For seven STEC strains confirmed by WGS, $stx$ was not located in regions where there were fragments of $stx$-encoded phage as determined by PHASTER pipeline (Table 3 and Fig 2). For five WGS-confirmed STEC strains, $stx$ was in the contig where $stx$-phage fragments were detected, with CAP 18 having both $stx_1$ and $stx_2$, but only $stx_1$ associated with phage DNA (Fig 3). The presence of $stx$ was verified near the insertion site of $NinF$ and $NinG$ genes in seven of the eleven STEC strains. However, $stx$ was located adjacent (within ten genes prior to $stx$ in the genome) to the $Lar$ family of genes in six of the STEC (Figs 2 and 3). A heatmap divided strains into 3 groups: (A) Fifteen STEC-negative strains by WGS lacking $stx$-phage insertions; (B) Twenty-two STEC-negative strains by WGS with $stx$-encoding phage insertions; and (C) eleven STEC-positive strains by WGS (Fig 4).

### 3.3 Subtypes of $stx$ and biofilm genes

All WGS-confirmed STEC strains possessed $stx_{1a}$ and $stx_{1b}$, with CAP03 and CAP18 also possessing $stx_{2a}$ and $stx_{2b}$ (Table 2). In all cases, if $stx_1$ and/or $stx_2$ were confirmed by WGS, both a and b subtypes were present and by extension two or four bacteriophages would have initially inserted $stx$ into these bacterial genomes. Biofilm genes detected by WGS included $csgB$, $csgD$, $csgE$, $csgF$, $csgG$ in all STEC, and (47/48) of all strains sequenced. Other genes including $cheY$, $entABCEFS$, $espX4$, $espX5$, $fepABCG$, $flgG$, and $ompA$ were present in all 48 $Escherichia$ $coli$

**Table 3. Presence of *stx*-encoding bacteriophages.**

| Strain | Phage | Integrity of phage in sequence (%) | Integrity of *stx* in phage sequence (%) | Position of phage in contig sequence | Description of *stx* associated with phage[2] | Position of *stx* in genome |
|---|---|---|---|---|---|---|
| CAP 01 | Enterobacteria phage 933W (NC_000924) | 60 | 1.8 | 25 | $stx_2$ missing | . |
| | Enterobacteria phage YYZ-2008 (NC_011356) | 4 | 38.7 | 36 | $stx_1$ fragment | |
| [1]CAP 02 | Enterobacteria phage BP-4795 (NC_004813) | 4 | 16.9 | 67 | $stx_1$ fragment | 109 |
| [1]CAP 03 | Enterobacteria phage YYZ-2008 (NC_011356) | 7 | 35.6 | 30 | $stx_1$ fragment | 90 |
| [1]CAP 04 | Enterobacteria phage BP-4795 (NC_004813) | 4 | 17.3 | 70 | $stx_1$ fragment | 83 |
| CAP 05 | Enterobacteria phage BP-4795 (NC_004813) | 20 | 28.3 | 15 | $stx_1$ fragment | . |
| CAP 06 | Enterobacteria phage BP-4795 (NC_004813) | 8 | 5.4 | 1 | $stx_1$ fragment | . |
| CAP 08 | Enterobacteria phage BP-4795 (NC_004813) | 4 | 7.3 | 62 | $stx_1$ fragment | . |
| | Enterobacteria phage BP-4795 (NC_004813) | 6 | 0.9 | 49 | $stx_1$ missing | |
| | Enterobacteria phage YYZ-2008 (NC_011356) | 5 | 0.0 | 1 | $stx_1$ missing | |
| [1]CAP 10 | Enterobacteria phage BP-4795 (NC_004813) | 12 | 18.0 | 10 | $stx_1$ fragment | 47 |
| | Enterobacteria phage Min27 (NC_010237) | 1 | 23.0 | 45 | $stx_2$ fragment | |
| | Enterobacteria phage YYZ-2008 (NC_011356) | 1 | 0.0 | 70 | $stx_1$ missing | |
| | Stx1 converting phage DNA (NC_004913) | 16 | 100 | 47 | $stx_{1a}$ and $stx_{1b}$ | |
| | *Shigella* phage POCJ13 (NC_025434) | 2 | 9.5 | 38 | $stx_1$ fragment | |
| CAP 11 | Enterobacteria phage BP-4795 (NC_004813) | 7 | 25.7 | 4 | $stx_1$ fragment | . |
| | Stx2 converting phage II DNA (NC_004914) | 11 | 16.1 | 18 | $stx_2$ fragment | |
| CAP 14 | Enterobacteria phage 933W (NC_000924) | 23 | 0.0 | 11 | $stx_2$ missing | . |
| | Enterobacteria phage YYZ-2008 (NC_011356) | 2 | 8.6 | 1 | $stx_1$ fragment | |
| | Escherichia phage PA28 (NC_041935) | 28 | 0.0 | 56 | $stx_2$ missing | |
| | Stx2-converting phage 1717 (NC_011357) | 4 | 10.7 | 20 | $stx_2$ fragment | |
| CAP 15 | Enterobacteria phage BP-4795 (NC_004813) | 32 | 0.0 | 47 | $stx_1$ missing | . |
| | Enterobacteria phage YYZ-2008 (NC_011356) | 1 | 23.3 | 15 | $stx_1$ fragment | |
| | Shigella phage POCJ13 (NC_025434) | 1 | 24.0 | 50 | $stx_1$ fragment | |
| | Shigella phage POCJ13 (NC_025434) | 1 | 17.0 | 7 | $stx_1$ fragment | |
| [1]CAP 16 | Enterobacteria phage BP-4795 (NC_004813) | 6 | 16.9 | 32 | $stx_1$ fragment | 79 |
| CAP 17 | Enterobacteria phage YYZ-2008 (NC_011356) | 3 | 16.5 | 9 | $stx_1$ fragment | . |

*(Continued)*

**Table 3.** (Continued)

| Strain | Phage | Integrity of phage in sequence (%) | Integrity of *stx* in phage sequence (%) | Position of phage in contig sequence | Description of *stx* associated with phage[2] | Position of *stx* in genome |
|---|---|---|---|---|---|---|
| [1]CAP 18 | Enterobacteria phage 933W (NC_000924) | 28 | 0.0 | 32 | $stx_2$ missing | 34 |
| | Enterobacteria phage YYZ-2008 (NC_011356) | 3 | 29.4 | 16 | $stx_1$ fragment | |
| | Escherichia phage PA28 (NC_041935) | 21 | 0.0 | 53 | $stx_2$ missing | |
| | Shigella phage Ss-VASD (NC_028685) | 12 | 99.8 | 34 | $stx_{1a}$ and $stx_{1b}$ | |
| | Shigella phage Ss-VASD (NC_028685) | 29 | 8.5 | 49 | $stx_1$ fragment | |
| | Shigella phage 75/02 Stx (NC_029120) | 2 | 9.6 | 50 | $stx_1$ fragment | |
| CAP 20 | Enterobacteria phage BP-4795 (NC_004813) | 12 | 14.0 | 10 | $stx_1$ fragment | . |
| CAP 22 | Enterobacteria phage BP-4795 (NC_004813) | 1 | 11.4 | 69 | $stx_1$ fragment | . |
| | Shigella phage POCJ13 (NC_025434) | 2 | 26.3 | 10 | $stx_1$ fragment | |
| [1]CAP 23 | Enterobacteria phage BP-4795 (NC_004813) | 29 | 100 | 24 | $stx_{1a}$ and $stx_{1b}$ | 24 |
| | Enterobacteria phage YYZ-2008 (NC_011356) | 3 | 29.5 | 16 | $stx_1$ fragment | |
| CAP 25 | Enterobacteria phage BP-4795 (NC_004813) | 10 | 0.6 | 4 | $stx_1$ missing | . |
| | Shigella phage 75/02 Stx (NC_029120) | 1 | 5.5 | 18 | $stx_1$ fragment | |
| CAP 26 | Enterobacteria phage BP-4795 (NC_004813) | 1 | 11.0 | 2 | $stx_1$ fragment | . |
| CAP 30 | Enterobacteria phage BP-4795 (NC_004813) | 8 | 7.9 | 2 | $stx_1$ fragment | . |
| [1]CAP 32 | Enterobacteria phage 933W (NC_000924) | 9 | 37.2 | 32 | $stx_2$ fragment | 48 |
| | Enterobacteria phage YYZ-2008 (NC_011356) | 1 | 0.0 | 14 | $stx_1$ missing | |
| | Escherichia phage PA28 (NC_041935) | 19 | 56.3 | 48 | $stx_{1a}$ and $stx_{1b}$ | |
| [1]CAP 33 | Enterobacteria phage 933W (NC_000924) | 18 | 1.0 | 63 | $stx_2$ missing | 105 |
| | Escherichia phage PA28 (NC_041935) | 21 | 19.0 | 66 | $stx_2$ fragment | |
| | Escherichia phage PA28 (NC_041935) | 1 | 0.0 | 57 | $stx_2$ missing | |
| CAP 34 | Enterobacteria phage YYZ-2008 (NC_011356) | 1 | 20.5 | 62 | $stx_1$ fragment | . |
| CAP 35 | Enterobacteria phage BP-4795 (NC_004813) | 7 | 24.1 | 16 | $stx_1$ fragment | . |
| CAP 39 | Enterobacteria phage BP-4795 (NC_004813) | 7 | 7.3 | 2 | $stx_1$ fragment | . |
| CAP 42 | Enterobacteria phage YYZ-2008 (NC_011356) | 1 | 3.0 | 1 | $stx_1$ fragment | . |
| | Shigella phage POCJ13 (NC_025434) | 2 | 19.3 | 9 | $stx_1$ fragment | |

(*Continued*)

**Table 3.** (Continued)

| Strain | Phage | Integrity of phage in sequence (%) | Integrity of $stx$ in phage sequence (%) | Position of phage in contig sequence | Description of $stx$ associated with phage[2] | Position of $stx$ in genome |
|---|---|---|---|---|---|---|
| CAP 44 | Enterobacteria phage YYZ-2008 (NC_011356) | 1 | 12.7 | 8 | $stx_1$ fragment | . |
| CAP 45 | Enterobacteria phage YYZ-2008 (NC_011356) | 3 | 17.2 | 7 | $stx1$ fragment | . |
| CAP 46 | Enterobacteria phage BP-4795 (NC_004813) | 16 | 25.7 | 6 | $stx1$ fragment | . |
| [1]CAP 47 | Enterobacteria phage BP-4795 (NC_004813) | 29 | 100 | 24 | $stx_{1a}$ and $stx_{1b}$ | 24 |
| | Enterobacteria phage YYZ-2008 (NC_011356) | 3 | 0.0 | 16 | $stx_1$ missing | |
| CAP 48 | Enterobacteria phage YYZ-2008 (NC_011356) | 1 | 10.7 | 2 | $stx_1$ fragment | . |
| CAP 49 | Enterobacteria phage BP-4795 (NC_004813) | 4 | 32.5 | 44 | $stx_1$ fragment | . |

[1]STEC confirmed by WGS. Fifteen isolates (CAP 9, 12, 13, 21, 24, 27, 28, 29, 31, 36, 37, 38, 40, 41, and 43) did not have sequences attributable to $stx$-encoding phage in their genome. CAP 19 was STEC as determined by WGS but did not show any $stx$-encoding phage.

[2] $stx$ missing: the phage integrated into the bacterial genome does not contain even a fragmentary $stx$. Fragment: presence of truncated $stx$ subunits in bacterial genome with some base pair mismatches.

strains, reinforcing the quality of the coverage of the sequencing of the isolates (S1 Table). Finally, other genes that regulate cell surface adhesins were verified, such as *FimA* and *FimB* (S1 Table).

### 3.5 Serogroup and serotype

For serogroup determination, PCR and WGS were in agreement for 29/50 isolates (Table 4). PCR and WGS fully agreed with the assignment of the 11 STEC strains to their O-groups. Therefore, all mismatches between PCR and WGS serogroup (21/50) were in generic *E. coli* isolates (non STEC by WGS). In summary, generic *E. coli* strains showed false positive amplification for serogroups: O26 (n = 6), O45 (n = 2), O103 (n = 6), O145 (n = 2), and O157 (n = 5). The exceptions were O121 which had stable serogroup detection (Table 4) and O111 which was not included in this study due to previously noted stable serogroup and $stx_1$ detection [33].

### 3.6 Resistome and plasmids

The *arsB-mob* gene which encodes resistance to arsenic was present in 7/11 STEC isolates and *BlaEC* which encodes for beta-lactamase resistance was present in all *E. coli* (Table 5). Other resistance genes to various antimicrobials were occasionally identified including aminoglycosides, diaminopyrimidines, sulfonamides, quaternary amines, tetracycline and phenols. Six generic *E. coli* strains (CAP 5, 21, 24, 29, 34, 39) carried three or more AMR genes. Almost all STEC isolates harbored at least one plasmid, with *IncFIB (AP001918)1* being the most common, and CAP47 the only STEC strain that lacked plasmids.

### 3.7 MLST and Phylogenomic relationship between strains

For all *E. coli* isolates, 29 sequence types (ST) were detected, but for STEC strains, only six STs were identified (11, 21, 32, 343, 723, and 5082; Table 6). For O157:H7, ST11 strains were detected, similar to that of the reference strain used (*Escherichia coli* O157:H7 str. Sakai DNA, sequence BA000007), emphasizing the potential pathogenicity of our strains.

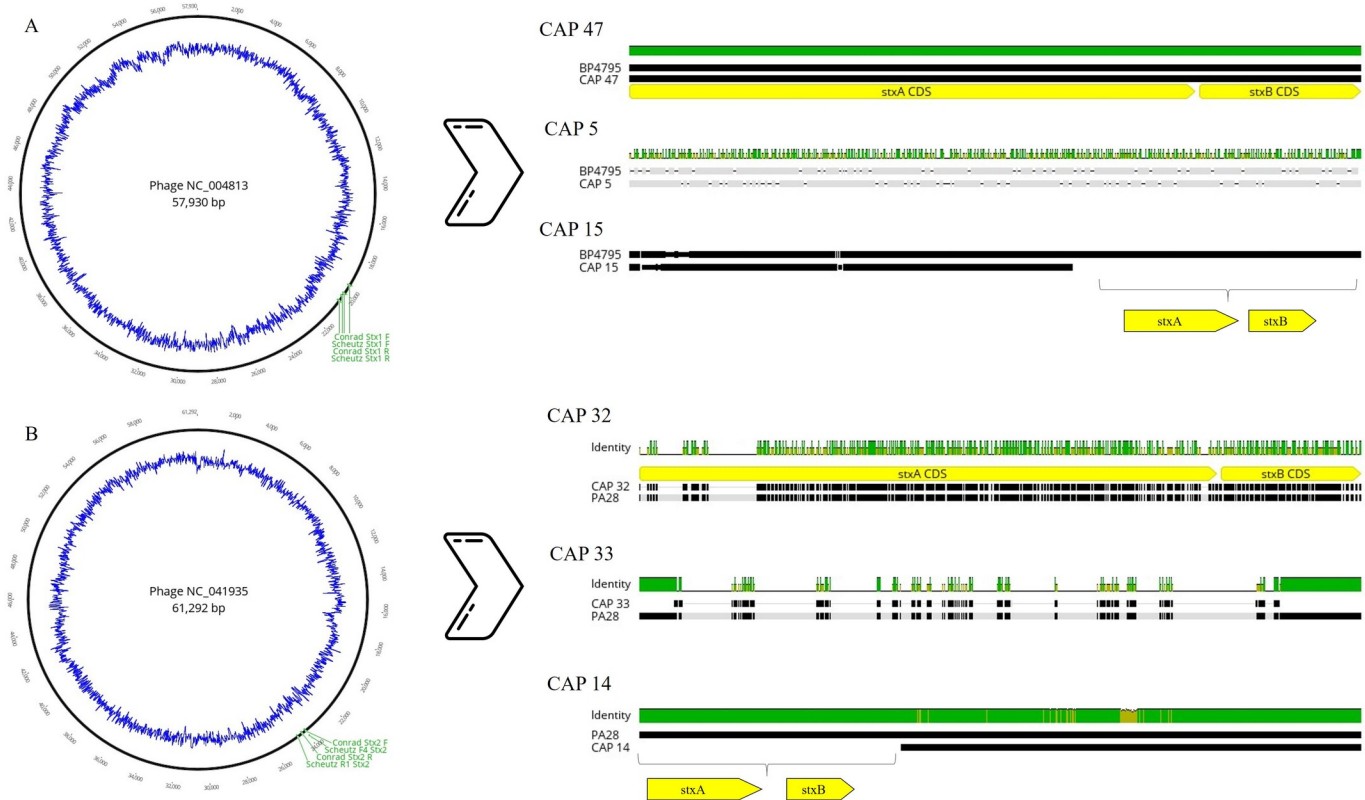

**Fig 1. The insertion of *stx*-encoding phage and complete *stxA* and *stxB* subunits in STEC (CAP 47, CAP 32), *stx* fragments (truncated *stxA* and *stxB* subunits with some base substitutions; CAP 5, CAP 33) and two non-STEC with inserted phage but lacking *stx* coding regions (CAP 15, CAP 14).** A: NC_004813 (*Enterobacteria* phage BP-4795). STEC: CAP 47. non-STEC: CAP 5, CAP 15; B: NC_041935 (*Escherichia* phage PA28). STEC: CAP 32 and CAP 33. non-STEC: CAP 14.

Based on ST results, O103:H11 may be more closely related to O26:H11 than to O103:H25. In addition, O145:H28 was closely related to O157:H7 as they both had the same subtypes of *stx* ($stx_{1a}$, $stx_{1b}$, $stx_{2a}$, $stx_{2b}$). A phylogenomic tree with 0.055 relatedness was developed using a single copy of each isolate plus the reference genome using multi-locus sequence types (Fig 5).

## 4. Discussion

### 4.1 Isolation of *Citrobacter* spp

*Citrobacter* spp. is part of the *Enterobacteriaceae* family and can grow in the enrichment medium of *Escherichia coli*, with morphology very similar to that of STEC colonies [34]. Using IMS may have also led to this misidentification, since some strains of *Citrobacter* spp. express an antigen similar to that of O157 [35]. Moreover, *Citrobacter* spp. strains positive for *stx* have been previously described [36]. A possible solution to prevent misidentification of *Citrobacter spp.* would be additional PCR assays to detect the *uidA* gene, responsible for the activity of beta-glucuronidase (mainly for O157), or housekeeping genes for *E. coli*, such as *arcA*, *gapA*, *mdh*, *rfbA*, and *rpoS* [37].

Possibly, amplification of a free *stx*-encoding phage may have occurred at initial isolation as the two *Citrobacter* isolated in the present study did not have *stx*-encoding phage fragments in their genomes. Other PCR-based studies of *E. coli* have also either detected free *stx*-encoding phages or hypothesized the loss of *stx* after sub-cultivation [13,16, 38]. Free *stx*-phages have been found in *Citrobacter* spp. [36] and other species such as *Escherichia albertii* [39].

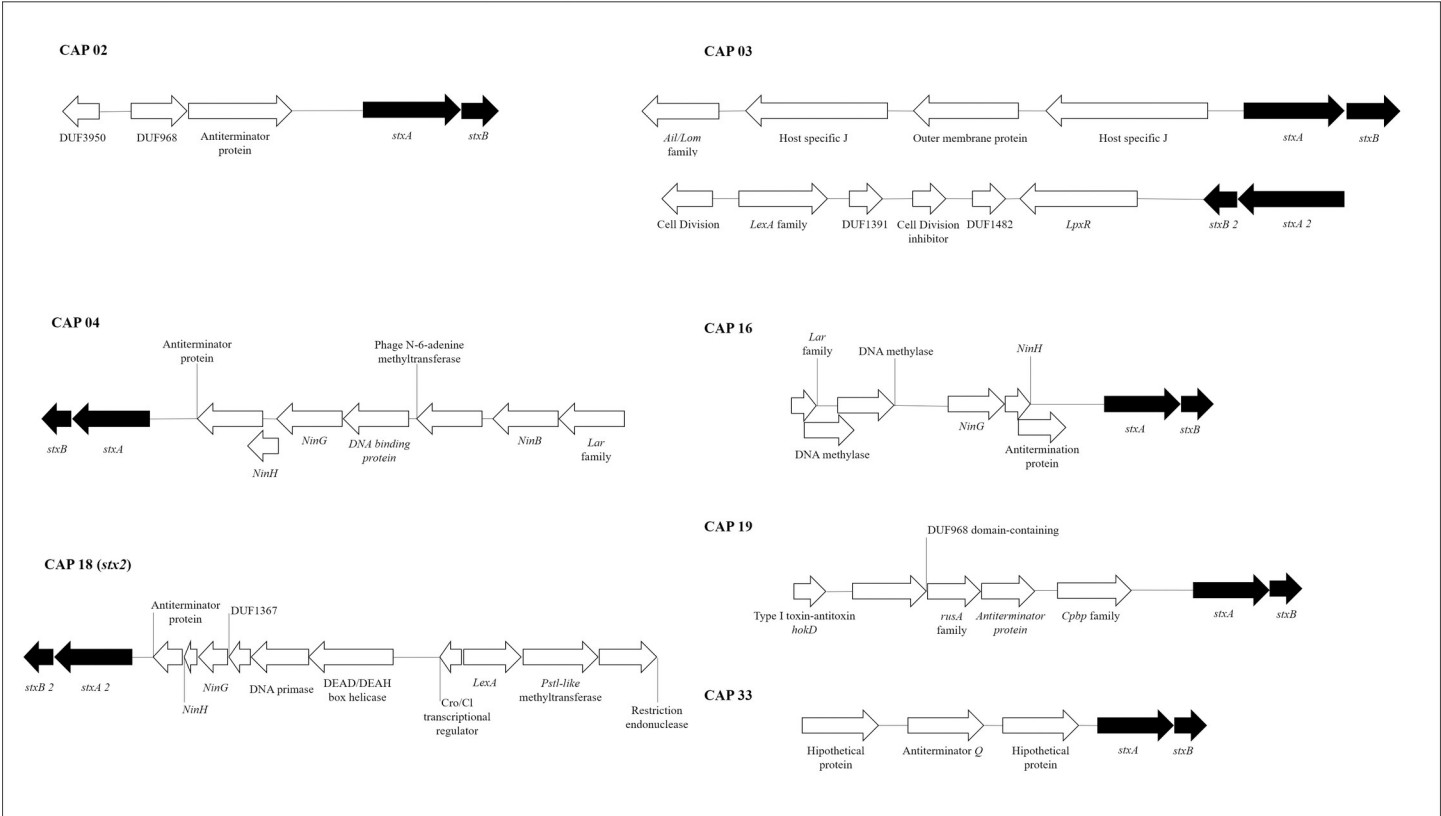

**Fig 2. Linear STEC sequences with *stx* insertion in a contig without *stx*-encoding phage determined by Phaster.** *StxA* and s*txB* subunits are shown in dark arrows for better visualization of the genome and neighboring genes.

Additional complicating factors which increase the difficulty of isolating STEC include adaptability (e.g. change in the expression of some genes) and difficulty in establishing a culture medium that can promote uniform growth between STEC strains [8]. Immunomagnetic separation was used to overcome some of the difficulties in isolating STEC in the present study. However, it is worth mentioning that as IMS is serogroup-based, it has a small spectrum of detection due to the large number of existing STEC serogroups [1)] Also, some cross reactions among serogroups have occurred, decreasing the discriminatory power of IMS [40, 41]. Other challenges in isolation of STEC were addressed by our group in previous studies [42, 43]. Competition through culture, the differences between detection across laboratories, and the lack of selectivity by IMS highlight the need to improve methodologies for detection and the isolation of STEC [8]. Consequently, the use of different culture media which would be selective for all STEC and/or the development of new IMS beads with increased selectivity would simplify STEC detection and isolation. Although WGS also has weaknesses with some inherent to the Illumina platform including decreased quality toward the ends of reads, non-uniform amplification of target regions, and difficulties in assembly due to the short length of sequences [44], a combination of phenotypic approaches aligned with genotypic tools can better guarantee effective STEC isolation in future studies.

## 4.2 Concordance of Shiga toxin genes by PCR and WGS and phage influence in PCR

Although there may be difficulties in isolation of STEC, and it has been established that PCR assays across laboratories can produce variable results for detection of Shiga toxin genes due

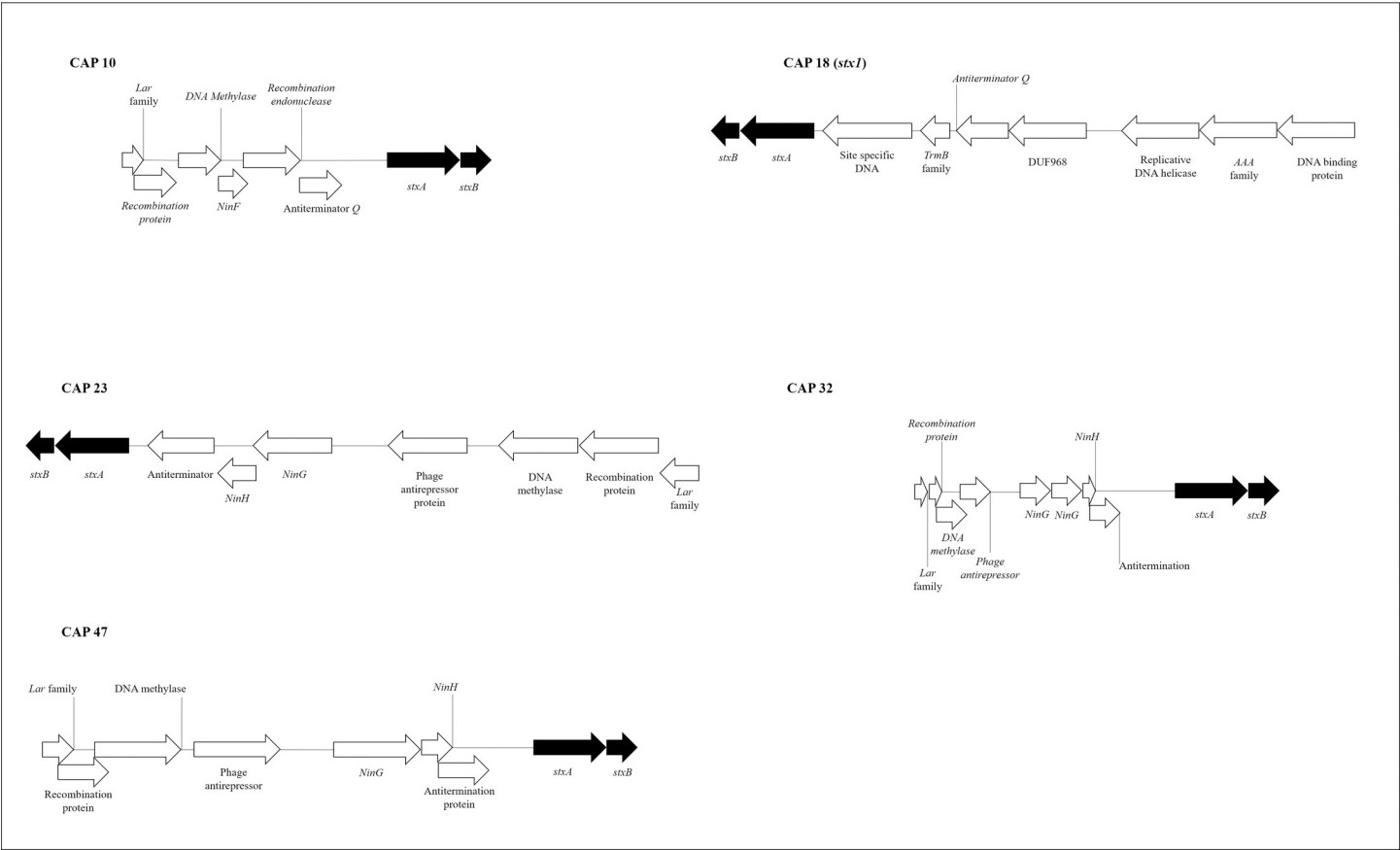

**Fig 3. Linear STEC sequence with *stx* insertion in a location of *stx*-encoding phage.** StxA and *stx*B subunits are shown in dark arrows for better visualization of the genome and neighboring genes.

use of different equipment and methods [22, 33], it was expected that the use of the same assay by the same laboratory staff with the same equipment and conditions would produce consistent results. However, on re-growth of isolates collected in previous studies, and repeated PCR, detection of $stx_1$ and/or $stx_2$ showed variation for some isolates. Fourteen isolates which were positive for *stx* in the first PCR were negative in the second assay, matching WGS results, although twenty-five isolates continued to show false-positive PCR results in the second assay (positive in PCR but negative in WGS; Table 2). Loss of *stx* genes after re-culture has been previously described [16] and may be also be attributed to mixed cultures (containing multiple strains of *E. coli* either possessing or lacking *stx*, resulting in variability depending on which colonies are selected) or loss of free *stx*-carrying phage [14].

*Stx* is carried by phages that may be free within the cell at the start of the lysogenic cycle prior to phage DNA insertion into the bacterial chromosome [13, 45]. Although there is great heterogeneity of phages encoding Shiga toxins, the location of phage insertion in the bacterial genome has been reported to be close to *wrbA* or *yecE* in the Q terminator region [46]. However, based on results of the present study, seven STEC strains instead had *stx* inserted close to *NinF* and *NinG*.

The adjacent gene relationship between *stx*-phage insertion and *NinG* has been previously reported in O157:H7, with *NinG* thought to act as a controller of *stx* expression [47]. As seven STEC strains had the insertion of *stx* near to *NinF* and *NinG*, it is possible that these strains had a greater *stx* stability in the genome and less likelihood of undergoing a phage excision process.

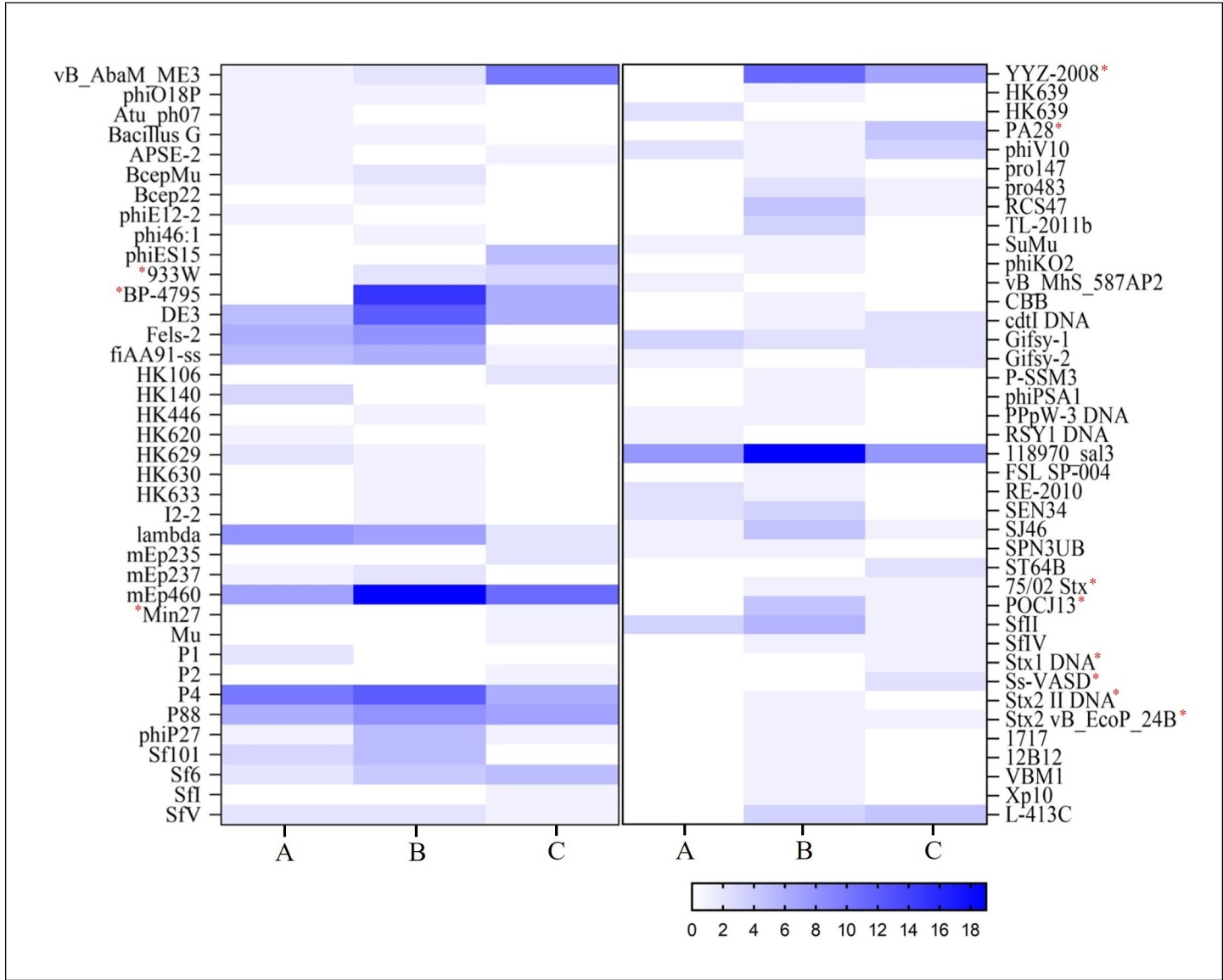

**Fig 4.** Heat map dividing strains used in the present study in 3 groups: (A) Fifteen STEC-negative strains by WGS lacking *stx*-phage fragments; (B) Twenty-two STEC-negative strains by WGS with *stx*-encoding phage fragments; (C) Eleven STEC-positive strains by WGS. A more intense blue color indicates that the phage sequence was more prevalent in that set of strains. Red asterisk identifies *stx*-phages.

Also, seven of eleven strains confirmed as STEC by WGS lacked phage DNA flanking *stx* insertion sites (Table 3 and Fig 2). The lack of detection of phage DNA may represent cryptic phage which have lost the ability to excise from the bacterial genome, similar to those carrying *stx₁* in *E. coli* O111 [48]. *Stx* is typically a single transcriptional unit consisting of A and B subunits [49], but multiple insertion, mutation and excision events may have led to defective *stx*-prophages, and these occurrences can be considered as pathoadaptive mutations, although it is not known what advantage the cell obtains from immobilizing *stx* [48]. Of interest, Creuzburg et al. [48] also obtained variable *stx* PCR results which were attributed to a lack of primer-binding sites, missing fragments of the target genes, or the presence of other mobile genetic elements causing PCR amplification.

Table 4. Comparison of serogroup between PCR and WGS[1].

| Strain | Serogroup by PCR | Serotype by WGS |
|---|---|---|
| CAP 01 | O103 | O103:H2 |
| CAP 02[2] | O103 | O103:H11 |
| CAP 03[2] | O157 | O157:H7 |
| CAP 04[2] | O26 | O26:H11 |
| CAP 05 | O26 | O9:H30 |
| CAP 06 | O45 | O110:H30 |
| CAP 07 | O26 | *Citrobacter* sp. |
| CAP 08 | O103 | O103:H2 |
| CAP 09 | O103 | H34 |
| CAP 10[2] | O157 | O157:H7 |
| CAP 11 | O121 | O121:H7 |
| CAP 12 | O45 | O9:H4 |
| CAP 13 | O26 | H28 |
| CAP 14 | O103 | O103:H2 |
| CAP 15 | O45 | O45:H51 |
| CAP 16[2] | O26 | O26:H11 |
| CAP 17 | O26 | O17:H18 |
| CAP 18[2] | O145 | O145:H28 |
| CAP 19[2] | O121 | O121:H7 |
| CAP 20 | O103 | O17:H18 |
| CAP 21 | O103 | O153:H8 |
| CAP 22 | O145 | O8:H2 |
| CAP 23[2] | O145 | O145:H28 |
| CAP 24 | O145 | O76:H34 |
| CAP 25 | O121 | O121:H7 |
| CAP 26 | O45 | O45:H11 |
| CAP 27 | O103 | O103:H8 |
| CAP 28 | O103 | O5:H32 |
| CAP 29 | O103 | O5:H19 |
| CAP 30 | O157 | H34 |
| CAP 31 | O157 | O157:H29 |
| CAP 32[2] | O145 | O145:H28 |
| CAP 33[2] | O103 | O103:H25 |
| CAP 34 | O26 | O8:H10 |
| CAP 35 | O45 | O45:H45 |
| CAP 36 | O26 | O26:H9 |
| CAP 37 | O103 | O187:H52 |
| CAP 38 | O157 | O157:H29 |
| CAP 39 | O45 | O45:H4 |
| CAP 40 | O157 | O53:H32 |
| CAP 41 | O103 | O103:H19 |
| CAP 42 | O26 | O26:H32 |
| CAP 43 | O157 | O51:H14 |
| CAP 44 | O45 | O45:H38 |
| **CAP 45** | O157 | O157:H12 |
| **CAP 46** | O103 | O103:H21 |
| CAP 47[2] | O145 | O145:H28 |

(*Continued*)

**Table 4.** (Continued)

| Strain | Serogroup by PCR | Serotype by WGS |
|--------|------------------|-----------------|
| CAP 48 | O26 | O157:H38 |
| CAP 49 | O157 | O103:H14 |
| CAP 50 | O157 | *Citrobacter* sp. |
| **Agreement** | **29/50** | |

[1]Grey shaded isolates showing agreement between PCR and WGS, yellow shaded isolates where inconclusive serotyping by WGS.

[2]Isolates confirmed as STEC by WGS.

Environments with a high bacterial density promote transfer of phages, with phages being both gained and lost by bacterial members within this dynamic environment [50]. In addition, the presence of multiple fragments of *stx*-coding phages may be related to the loss of phages by sub-cultivation, which has already been demonstrated [15, 16, 38, 51]. Based on our results, we would agree with Senthakumaran et al. [38] who concluded that STEC with intact prophages may be uncommon and difficult to detect. Also, using WGS these authors observed the existence of a *stx*-negative "*in vivo*" strain O145:H28 with characteristics similar to another STEC strain of the same serotype [38]. Moreover, studies evaluating *stx* loss suggest that STEC O157:H7 strains are more "*stx* stable" when compared to non-O157 serogroups [16, 38, 51], although our study also included O157 strains selected for *stx* instability (n = 7). However, a difference between the present study and other studies that evaluated the loss of *stx* phage is that in our results the loss of $stx_1$ was more common, likely due to its increased prevalence, while other studies investigated the loss of $stx_2$ [16, 38, 51].

A significant finding of the present study was that intermittent false *stx* positives could in twenty-two cases be possibly related to presence of fragments of *stx*-encoding phages (Table 2), especially as genomes of the majority of strains possessed multiple fragments of identifiable *stx*-encoding phages (Table 3). The Conrad et al. [10] primers used at initial isolation have had positive amplification of *stx* even with one or two base-pair mismatches [33], but the possible intermittent binding of primers to *stx* fragments has not been previously reported, likely as only a subset of *stx* fragments may have influenced PCR results. Larger fragments with highest *stx* sequence integrity would be the most likely to intermittently bind to PCR primers, although it was not possible in the present study to verify which if any of the *stx* fragments led to false-positive PCRs. However, it is likely more than coincidence that all isolates with fragments having at least 23% $stx_1$ or $stx_2$ integrity (n = 9) had intermittent PCR detection of that gene unless they also had an intact *stx* of the same type enabling consistent PCR detection. The $stx_1$ present in CAP 32 is interesting and possibly intermediate to a fragment and a complete *stx* as it only had 56% $stx_1$ integrity in Geneious analyses due to base substitutions, but was classified as STEC by WGS. Accordingly, the demarcation between STEC and non-STEC may be more complicated than previously supposed and investigating expression of Shiga toxins would provide further clarity.

Three types of insertion of *stx*-encoding phage in the bacterial genome were verified (Fig 1). The CAP 47 strain confirmed as STEC showed homology with *stx*-encoding phage BP 4795, while two other non-STEC strains had multiple insertions between the bases of the *stx*-phage encoding region (CAP 5, CAP 33). In contrast, CAP 14 and CAP 15 each had a conserved *stx*-carrying phage in their genome but lacked a *stx* coding region. Similar to CAP 14 and CAP 15 strains, Senthakumaran et al. [38] noted the absence of *stx* in strains

**Table 5. Presence of resistance genes and plasmids in *E. coli* and STEC.**

| Strains | Resistance genes | Plasmids by WGS |
|---|---|---|
| CAP01 | *arsB-mob; blaEC-18* | IncFIB(AP001918)_1 |
| CAP02[1] | *arsB-mob; blaEC-18* | ColRNAI_1; IncB/O/K/Z_3; IncFIB(AP001918)_1 |
| CAP03[1] | *arsB-mob; blaEC-15* | IncFIA_1; IncFIB(AP001918)_1 |
| CAP04[1] | *arsB-mob; blaEC-18* | ColRNAI_1; IncB/O/K/Z_3; IncFIB(AP001918)_1; p0111_1 |
| CAP05 | *aph(3")-Ib; aph(6)-Id; arsB-mob; blaEC-15; blaTEM-1; dfrA5; sul2* | IncFIB(AP001918)_1; IncFII_1; IncQ1_1; IncX1_1; IncX3_1 |
| CAP06 | *arsB-mob; blaEC-13; tet(A)* | Col156_1; IncFIB(AP001918)_1; IncFIC(FII)_1; IncI1_1_Alpha; IncY_1 |
| CAP08 | *arsB-mob; blaEC-18* | IncFIC(FII)_1; IncY_1 |
| CAP09 | *arsB-mob; blaEC* | ColRNAI_1; IncFIA(HI1)_1_HI1; IncFIB(K)_1_Kpn3 |
| CAP10[1] | *arsB-mob; blaEC-15* | IncFIB(AP001918)_1 |
| CAP11 | *arsB-mob; blaEC-18* | - |
| CAP12 | *arsB-mob; blaEC-18* | - |
| CAP13 | *blaEC* | IncFIA_1; IncFIB(AP001918)_1; IncX1_1; IncX3_1 |
| CAP14 | *arsB-mob; blaEC-18* | IncFIA(HI1)_1_HI1; IncFII(pRSB107)_1_pRSB107; IncX1_1 |
| CAP15 | *arsB-mob; blaEC-18; tet(C)* | IncFIB(AP001918)_1; IncFII_1 |
| CAP16[1] | *arsB-mob; blaEC-18* | ColRNAI_1; IncB/O/K/Z_3; IncFIB(AP001918)_1 |
| CAP17 | *aph(3")-Ib; aph(6)-Id; arsB-mob; blaEC-8; tet(B)* | IncFIB(AP001918)_1; IncX1_1; IncX3_1 |
| CAP18[1] | *blaEC* | IncB/O/K/Z_3; IncFIB(AP001918)_1 |
| CAP19[1] | *arsB-mob; blaEC-18* | IncFIA_1; IncFIB(AP001918)_1; IncFIC(FII)_1; IncY_1 |
| CAP20 | *arsB-mob; blaEC-8* | IncFII(pCoo)_1_pCoo; IncI1_1_Alpha |
| CAP21 | *arsB-mob; blaEC-18; qacG2; tet(A); tet(M)* | Col156_1; ColRNAI_1; IncFII_1 |
| CAP22 | *arsB-mob; blaEC-18;* | ColpVC_1; IncFIA(HI1)_1_HI1; IncFIB(AP001918)_1; IncFIC(FII)_1; p0111_1 |
| CAP23[1] | *blaEC* | IncB/O/K/Z_3; IncFIB(AP001918)_1 |
| CAP24 | *aadA2; blaEC; dfrA12; qacEdelta1; sul1; tet(A)* | IncFIB(K)_1_Kpn3; IncR_1; IncY_1 |
| CAP25 | *arsB-mob; blaEC-18* | IncFIA_1; IncFIB(AP001918)_1; IncFIC(FII)_ |
| CAP26 | *arsB-mob; blaEC-13;* | IncFIB(AP001918)_1 |
| CAP27 | *arsB-mob; blaEC-18;* | IncFIC(FII)_1 |
| CAP28 | *arsB-mob; blaEC-15; tet(B)* | ColRNAI_1; IncFIA_1; IncFII_1; IncI1_1_Alpha; IncX4_1; p0111_1 |
| CAP29 | *arsB-mob; blaEC-18; qacG2; tet(A); tet(M)* | ColRNAI_1; ColRNAI_1; IncFII_1 |
| CAP30 | *arsB-mob; blaEC-15* | - |
| CAP31 | *arsB-mob; blaEC-15; tet(C)* | Col156_1; ColE10_1; IncFIA_1; IncFIB(AP001918)_1; IncFIC(FII)_1; IncX4_2 |
| CAP32[1] | *blaEC* | IncB/O/K/Z_3; IncFIB(AP001918)_1 |
| CAP33[1] | *arsB-mob; blaEC-18* | IncFIB(AP001918)_1 |
| CAP34 | *aadA1; aph(3")-Ib; aph(6)-Id; arsB-mob; blaEC; blaTEM-1; floR; sul2* | IncA/C2_1; IncI1_1_Alpha |
| CAP35 | *arsB-mob; blaEC-15; tet(C)* | Col(MG828)_1; IncFIA_1; IncFIB(AP001918)_1; IncX1_1; IncX3_1; IncY_1 |
| CAP36 | *arsB-mob; blaEC* | ColRNAI_1; IncFIA(HI1)_1_HI1; IncFIB(K)_1_Kpn3 |
| CAP37 | *arsB-mob; blaEC-18* | ColRNAI_1; IncFIA_1; IncFIB(AP001918)_1 |
| CAP38 | *arsB-mob; blaEC-15; tet(A)* | Col(MG828)_1; IncFIA_1; IncFIB(AP001918)_1; IncFIC(FII)_1; IncI1_1_Alpha; IncX1_4; IncX3_1 |
| CAP39 | *aph(3")-Ib; aph(6)-Id; arsB-mob; blaEC-18; floR; sul2; tet(A)* | IncA/C2_1 |
| CAP40 | *arsB-mob; blaEC-15;* | - |

*(Continued)*

**Table 5.** (Continued)

| Strains | Resistance genes | Plasmids by WGS |
|---------|------------------|-----------------|
| CAP41 | *arsB-mob; blaEC-18* | ColRNAI_1 |
| CAP42 | *arsB-mob; blaEC* | ColRNAI_1 |
| CAP43 | *arsB-mob; blaEC* | Col(MG828)_1; ColRNAI_1; IncFIA_1; IncX1_1; IncX3_1 |
| CAP44 | *arsB-mob; blaEC-18;* | IncFIA_1; IncFIB(AP001918)_1 |
| CAP45 | *arsB-mob; blaEC* | ColRNAI_1 |
| CAP46 | *arsB-mob; blaEC-18* | IncFIA(HI1)_1_HI1; IncFIB(pB171)_1_pB171 |
| CAP47[1] | *blaEC* | - |
| CAP48 | *arsB-mob; blaEC-18* | IncFIA_1; IncFIB(AP001918)_1 |
| CAP49 | *arsB-mob; blaEC-18* | IncFIC(FII)_1; IncI1_1_Alpha |

[1]Strains confirmed as Shiga toxin-producing *E. coli*.

with conserved regions of *stx*-encoding phage. The presence of inconsistencies between bases present and phage sequences suggests that mutations may have occurred over time. Similarly, we also found a conserved PA28 phage region in CAP 32 strain encoded $stx_1$ instead of the more usual $stx_2$ [52].

## 4.3 Subtypes of *stx*, biofilm genes

In a study of 444 isolates of O157 from human disease outbreaks, multiple copies of $stx_1$ and/or $stx_2$ occurred in 68% of isolates [53]. However, it is odd that only multiple copies of $stx_1$ or $stx_2$ were present in all STEC isolates in the present study which were selected for WGS due to consistent *stx* PCR results. Accordingly, we hypothesize that multiple bacteriophage insertions may increase *stx* stability within the *E. coli* genome. Similarly, it was two STEC that had the highest number (five and six, respectively) of integrated *stx*-phage.

Almost all biofilm genes identified were members of the *csg* family (unique exception was CAP 34; S1 Table). Genes from the *csg* family play an important role in regulating biofilm genes in *E. coli* [54]. These genes are responsible for the formation of curli, an extracellular proteinaceous fiber which is involved in binding of surfaces and cell-to-cell contact, also influencing host colonization [55]. Strains of O157 that express curli are thought to have an exacerbated production linked to a high capacity for biofilm formation [56]. Potentially, STEC expressing curli may be linked to the phenomenon of super-shedding ($>10^4$ cells/g of feces), which has been theorized to be due to formation of an intestinal biofilm that when periodically sloughed leads to high numbers of fecal STEC [57]. However, presence of csg genes does not guarantee biofilm formation by STEC [58] and evaluation of biofilm forming phenotypes would require further study.

## 4.4 Serogroup and serotype

O-antigen serogroups represent the outermost part of the lipopolysaccharide layer and currently for *Escherichia coli* there are 184 O-serogroups [59]. Recently, some studies have standardized PCR assays to determine both O-antigen polysaccharide [59] and H-flagellum [60] as serological tests are laborious and may cross-react with other serogroups [61]. In the present study we found that in generic *E. coli* strains (without *stx* presence by WGS) there were 18 strains mistakenly amplified as belonging to the "Top Seven" (Table 4). There were also three strains which could not be O-serogrouped by WGS, illustrating limitations also of WGS.

**Table 6. Multilocus sequence typing profiles (MLST) of the *E. coli* isolates.**

| Strains[1] | ST | Allele[3] | | | | | | |
|---|---|---|---|---|---|---|---|---|
| | | *adk* | *fumC* | *gyrB* | *icd* | *mdh* | *purA* | *recA* |
| CAP01 | 17 | 6 | 4 | 3 | 17 | 7 | 7 | 6 |
| **CAP02**[2] | 723 | 16 | 154 | 12 | 16 | 9 | 7 | 7 |
| **CAP03**[2] | 11 | 12 | 12 | 8 | 12 | 15 | 2 | 2 |
| **CAP04**[2] | 21 | 16 | 4 | 12 | 16 | 9 | 7 | 7 |
| CAP05 | 540 | 6 | 7 | 57 | 1 | 8 | 8 | 2 |
| CAP06 | 187 | 6 | 69 | 4 | 16 | 9 | 13 | 7 |
| CAP08 | 17 | 6 | 4 | 3 | 17 | 7 | 7 | 6 |
| CAP09 | 8076 | 204 | 1109 | 4 | 1 | 8 | 8 | 2 |
| **CAP10**[2] | 11 | 12 | 12 | 8 | 12 | 15 | 2 | 2 |
| CAP11 | 1610 | 6 | 4 | 3 | 18 | 9 | 8 | 2 |
| CAP12 | 46 | 8 | 7 | 1 | 8 | 8 | 8 | 6 |
| CAP13 | 1300 | 12 | 136 | 199 | 30 | 24 | 2 | 17 |
| CAP14 | 17 | 6 | 4 | 3 | 17 | 7 | 7 | 6 |
| CAP15 | 20 | 6 | 4 | 3 | 18 | 7 | 7 | 6 |
| **CAP16**[2] | 21 | 16 | 4 | 12 | 16 | 9 | 7 | 7 |
| CAP17 | 69 | 21 | 35 | 27 | 6 | 5 | 5 | 4 |
| **CAP18**[2] | 32 | 19 | 23 | 18 | 24 | 21 | 2 | 16 |
| **CAP19**[2] | 5082 | 6 | 4 | 3 | 18 | 11 | 8 | 2 |
| CAP20 | 69 | 21 | 35 | 27 | 6 | 5 | 5 | 4 |
| CAP21 | 109 | 6 | 6 | 1 | 16 | 9 | 13 | 2 |
| CAP22 | 392 | 6 | 6 | 14 | 18 | 7 | 7 | 71 |
| **CAP23**[2] | 32 | 19 | 23 | 18 | 24 | 21 | 2 | 16 |
| CAP24 | 1415 | 204 | 11 | 4 | 1 | 8 | 8 | 2 |
| CAP25 | 5082 | 6 | 4 | 3 | 18 | 11 | 8 | 2 |
| CAP27 | 13 | 6 | 6 | 5 | 9 | 9 | 8 | 2 |
| CAP28 | 10 | 10 | 11 | 4 | 8 | 8 | 8 | 2 |
| CAP29 | 109 | 6 | 6 | 1 | 16 | 9 | 13 | 2 |
| CAP31 | 515 | 57 | 11 | 1 | 109 | 7 | 8 | 2 |
| **CAP32**[2] | 32 | 19 | 23 | 18 | 24 | 21 | 2 | 16 |
| **CAP33**[2] | 343 | 77 | 7 | 7 | 18 | 65 | 56 | 7 |
| CAP34 | 1122 | 8 | 11 | 57 | 1 | 7 | 18 | 6 |
| CAP35 | 10 | 10 | 11 | 4 | 8 | 8 | 8 | 2 |
| CAP37 | 1248 | 6 | 29 | 12 | 1 | 9 | 8 | 7 |
| CAP39 | 336 | 9 | 4 | 33 | 18 | 11 | 8 | 6 |
| CAP40 | 10 | 10 | 11 | 4 | 8 | 8 | 8 | 2 |
| CAP41 | 755 | 6 | 23 | 15 | 18 | 9 | 12 | 7 |
| CAP42 | 10 | 10 | 11 | 4 | 8 | 8 | 8 | 2 |
| CAP43 | 1406 | 46 | 156 | 2 | 25 | 5 | 16 | 19 |
| CAP44 | 154 | 6 | 6 | 5 | 10 | 9 | 8 | 6 |
| CAP45 | 10 | 10 | 11 | 4 | 8 | 8 | 8 | 2 |
| CAP46 | 446 | 6 | 19 | 3 | 26 | 11 | 8 | 6 |
| **CAP47**[2] | 32 | 19 | 23 | 18 | 24 | 21 | 2 | 16 |
| CAP48 | 1113 | 6 | 6 | 12 | 10 | 9 | 8 | 7 |
| CAP49 | 8935 | 6 | 8 | 32 | 159 | 9 | 23 | 381 |

[1]MLST analysis did not result in known Sequence Types for strains CAP26, CAP30, CAP36 AND CAP38.

[2]Strains confirmed as Shiga toxin-producing *E. coli*.

[3] Housekeeping, single-copy genes used to determine the allelic profile or sequence type (ST).

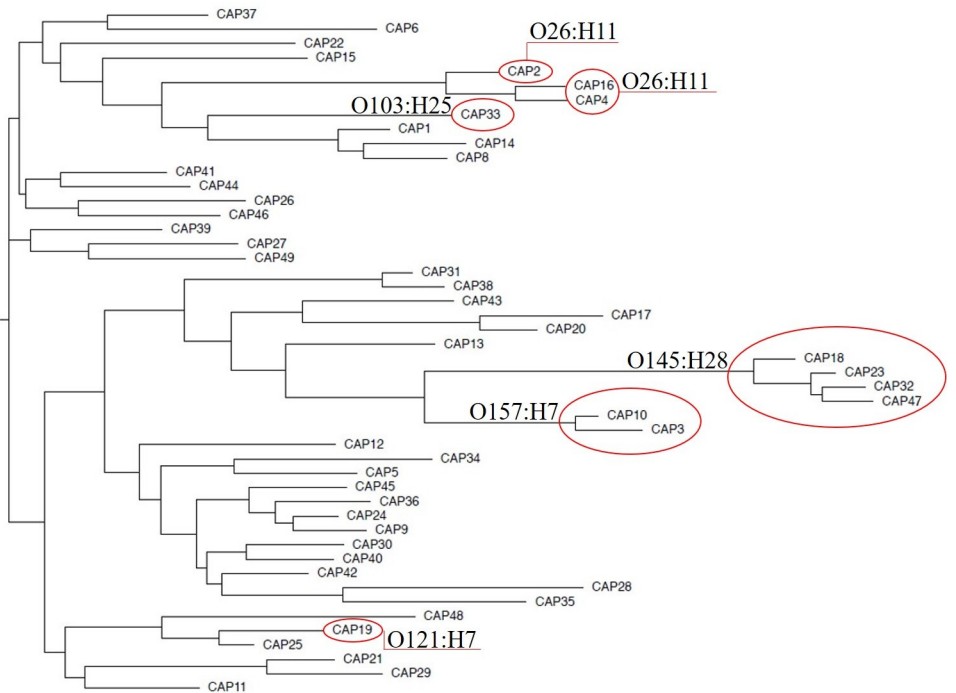

**Fig 5. Phylogenetic tree of the strains using the single nucleotide polymorphism (SNP) difference profile.**
*Escherichia coli* O157: H7 str. Sakai BA000007 was used as a DNA reference genome to build the SNP phylogenetic tree. The relatedness was calculated as 0.055. The red circles indicate proximity to a known ST outbreak strain.

For generic *E. coli* strains where serogroup determined by PCR did not match WGS, we evaluated whether there was a lack of primer specificity via virtual PCRs, and all primers evaluated only aligned with target regions. Also, all phages detected were evaluated by virtual PCR and did not affect possible amplification during serogroup determination. Therefore, our results emphasize that although the PCR for the determination of serogroup in STEC strains confirmed by WGS obtained 100% specificity, reasons for serogroup mismatches in some generic *E. coli* strains could not be determined. Mixed cultures are a possibility but unlikely to be wholly responsible. Additional study of unstable serogroup determination by PCR is required.

## 4.5 Resistome and plasmids

Information about antimicrobial resistance is important as antimicrobials are often included in media to improve the specificity of isolation methodologies. A number of antimicrobials including cefixime, cefsulodin, and vancomycin are used in enrichment broth for isolation of serogroup O157 [62]. Although arsenic and β-lactam resistance genes were present in most STEC strains in the present study, their use in culture media would not completely differentiate STEC from other *E. coli* strains due to the presence of these genes also in generic *E. coli* strains. However as selective media encompassing all STEC do not currently exist, the utility of β-lactam supplemented media is worthy of future exploration. The toxicity of arsenic would likely limit its practical application in culture media.

In relation to plasmid presence, *IncF* plasmids have been reported to confer resistance to different antimicrobials including β-lactams, aminoglycosides, tetracyclines, chloramphenicol, and quinolones [63, 64]. This plasmid is present in the class Inc that are responsible for producing TEM-1 or inhibitor-resistant TEM [65]. Moreover, *IncF* plasmids are widely

distributed in the *Enterobacteriaceae* family and contribute to the spread of antimicrobial multi-resistance among *E. coli* [66]. However, this plasmid class does not carry *stx* genes and would not have influenced PCR detection of $stx_1$ or $stx_2$.

## 4.6 Analysis of MLST profiles of strains

The multilocus sequence type of CAP03, ST11, has also been detected in cases of diarrhea as described by Ferdous et al. [67], in the database of the Food and Drug Administration from 2010 to 2017 [68], and confirmed in asymptomatic food handlers and from fecal sources of patients in Japan [69]. An important point is that O157:H7 is considered the serotype with the highest risk to humans, due to the large outbreaks that occurred in USA in 1993 [70, 71], in Japan in 1996 [72], and in Canada [73]. For this reason, the presence of O157:H7, and the ST11 profile, represents a direct risk of sporadic cases or a foodborne outbreak. Additionally, four isolates of ST32 (O145:H28) were detected. That ST is related to cases of hemolytic uremic syndrome. Furthermore, Shridhar et al. [74] analyzed 89 isolates of STEC serogroup O145 from several origins and all were ST32 with $stx_{1a}$ and $stx_{2a}$. However, in the present study, CAP18 also showed the presence of $stx_{1b}$ and $stx_{2b}$, which is evidence that supports the potential pathogenicity of this strain.

For serogroup O26:H11, two ST21 isolates were detected. This ST was detected in contamination from cattle feces [68], and hospitalized patients [75]. In addition, this ST was related to an outbreak occurring in Romania in 2016 where ST21 strains were isolated from 10 hemolytic uremic syndrome patients and five diarrhea cases [76]. Also, in a study by Chase-Topping et al. [77] which evaluated *E. coli* O26 isolated in Scottish cattle, ST21 was the most prevalent, but different from our strains, $stx_2$ was most common while only $stx_1$ was verified in our study.

The presence of ST343 (O103:H25) was described by Iguchi et al. [78] in sporadic cases and an outbreak with bloody diarrhea, vomiting and fever in Japan, and similar to present study, $stx_1$ was detected. In addition, this ST was isolated in areas of fish slaughter and watersheds [79]. As the strains isolated in our study were present in feces and animal hides, it is possible that they could also be present in water [80, 81].

For O103:H11 a ST723 was detected. Iguchi et al. [78] observed that serogroup O103 can be present in four ST groups [17, 343, 21, and 723]. The ST depends on the evolutionary line of each O103 strain. For example, ST723 is closely related to ST21, which in the present study was associated with an O26:H11. However, Eichhorn et al. [82], found that ST723 was related to isolates from humans, while ST21 was most often found in isolates from cattle. ST343 has a low similarity with ST21 and ST723, indicating a different evolution from the other two O103 sequence types.

Another ST, 5082, was detected for O121:H7. ST5082 is not common but was related to one bovine isolate and one of unknown origin in California [83]. In this same study, 85% of O121 serogroup isolated were ST655 and only 5% ST5082, but different to the present study had $stx_{1d}$ and $stx_{2a}$ or $stx_{1d}$ and $stx_{2c}$, while our strain carried $stx_{1a}$ and $stx_{1b}$. This divergence highlights the complexity and the ability for genetic rearrangement between strains of *E. coli*.

## 5. Conclusions

Generally, PCR is a reliable technique for classifying STEC and the few exceptions from our culture collection which had variable detection of *stx* and/or serogroup were investigated using WGS. In some cases, PCR primers used to determine *stx* genes may have been influenced by free phage encoding a Shiga toxin, since 29.2% of isolates (14/48) had concordant WGS and PCR results only in a second PCR after re-culture of the isolates. Conserved *stx*-encoding phages remaining in the genome without *stx* corroborates the possibility of loss in the region that encoded the *stx* gene, either by sub-cultivation or other unclear function. The presence of fragments of *stx* remaining in the genome may in some cases, particularly with

larger fragments, have led to intermittent amplification of PCR primers. Comparing serogroup among *E. coli* isolates as determined by PCR and WGS, both techniques agreed for STEC and in 18 generic *E. coli*, but in another 21 generic *E. coli* reasons for this incongruence could not be determined. It is unlikely that any technique may perfectly characterize STEC, but it is most important that Shiga toxin genes be reliably detected by PCR due to their potential human health risks. Having up to six integrated *stx*-phages per isolate including some lacking *stx*-coding regions and an average phage integrity of $< 10\%$ points to the extreme plasticity and impermanence of *stx*-carrying phage in the *E. coli* genome. Conversely, the majority of STEC lacked phage sequences in the same contig as *stx*, likely increasing stability of *stx* in the genome and its detection by PCR.

All STEC strains showed genes related to virulence, antimicrobial resistance, and adhesion to surfaces (biofilm formation), and when we analyzed the differences between the STEC isolates it was possible to verify that the main differences among isolates of the same serogroup were linked to the host cell-binding system. Strains showed a diversity of antimicrobial resistance genes, but all strains had a resistance gene for β-lactams. Consequently, β-lactams could be useful to improve isolation of STEC by inhibiting non-resistant background microflora. Regardless of difficulties in PCR classification, results of ST show a relation to other ST strains involved in food-borne outbreaks in other regions of the world, emphasizing the importance of accurate prediction of food safety risks.

## Supporting information

**S1 Table. Virulence genes detected in all STEC *E. coli* sequences.**
(DOCX)

## Acknowledgments

Many thanks to Ashwin Deo, Yidong Graham, Susanne Trapp, and Homayoun Zahiroddini for technical assistance.

## Author Contributions

**Conceptualization:** Kim Stanford.

**Formal analysis:** Vinicius Silva Castro, Rodrigo Ortega Polo, Eduardo Eustáquio de Souza Figueiredo, Robin King.

**Funding acquisition:** Eduardo Eustáquio de Souza Figueiredo, Carlos Adam Conte-Junior, Kim Stanford.

**Investigation:** Vinicius Silva Castro.

**Methodology:** Emmanuel Wihkochombom Bumunange.

**Project administration:** Kim Stanford.

**Resources:** Robin King, Carlos Adam Conte-Junior.

**Validation:** Rodrigo Ortega Polo.

**Writing – original draft:** Vinicius Silva Castro.

**Writing – review & editing:** Vinicius Silva Castro, Rodrigo Ortega Polo, Eduardo Eustáquio de Souza Figueiredo, Emmanuel Wihkochombom Bumunange, Tim McAllister, Robin King, Kim Stanford.

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
