## [Decision Letter · Decision Letter 0]

25 Aug 2021

Inconsistent PCR detection of Shiga toxin-producing Escherichia coli: insights from whole genome sequence analyses

PONE-D-21-16701

Dear Dr. Stanford,

We’re pleased to inform you that your manuscript has been judged scientifically suitable for publication and will be formally accepted for publication once it meets all outstanding technical requirements.

Kind regards,

Chitrita DebRoy

Academic Editor

PLOS ONE

Reviewers' comments:

Reviewer's Responses to Questions

**Comments to the Author**

1. Is the manuscript technically sound, and do the data support the conclusions?

Reviewer #1: Yes

Reviewer #2: Yes

2. Has the statistical analysis been performed appropriately and rigorously? 

Reviewer #1: N/A

Reviewer #2: Yes

3. Have the authors made all data underlying the findings in their manuscript fully available?

Reviewer #1: Yes

Reviewer #2: Yes

4. Is the manuscript presented in an intelligible fashion and written in standard English?

Reviewer #1: Yes

Reviewer #2: Yes

5. Review Comments to the Author

Reviewer #1: Overall a very concise and well written research article. The authors have used a good set of samples to test using both methods that they are comparing. The data clearly indicates what the authors are trying to report through this study that PCR detection of shiga toxin genes can be inconsistent using PCR.

Reviewer #2: The paper is well written and describes the causes of inconsistent detection of virulence genes, Shiga toxins, in Shiga toxin producing E. coli (STEC). By using whole genome sequencing they have studied in depth the causes of inconsistency, mainly due to the presence of phage DNA. The investigation is thorough and the work may lead to better understanding of the role phage DNA plays in STEC strains and measures can be developed to mitigate inconsistent detection of STEC by PCR for food safety.

6. PLOS authors have the option to publish the peer review history of their article (what does this mean?). If published, this will include your full peer review and any attached files.

Reviewer #1: No

Reviewer #2: No

---

## [Editor Report · Acceptance letter]

27 Aug 2021

PONE-D-21-16701 

*Inconsistent PCR detection of Shiga toxin-producing Escherichia coli: insights from whole genome sequence analyses*  

Dear Dr. Stanford:

I'm pleased to inform you that your manuscript has been deemed suitable for publication in PLOS ONE. Congratulations! Your manuscript is now with our production department. 

Kind regards, 

on behalf of

Dr. Chitrita DebRoy 

%CORR_ED_EDITOR_ROLE%

PLOS ONE